# Novel participatory methods for co-building an agent-based model of physical activity with youth

Leah Frerichs[1,2]*, Natalie Smith[1,3], Jill A. Kuhlberg[1,4], Gretchen Mason[1], Damie Jackson-Diop[5], Doris Stith[6], Giselle Corbie-Smith[2,7,8], Kristen Hassmiller Lich[1]

**1** Department of Health Policy and Management, Gillings School of Public Health, University of North Carolina at Chapel Hill, Chapel Hill, NC, United States of America, **2** Department of Social Medicine, Center for Health Equity Research, University of North Carolina at Chapel Hill, Chapel Hill, NC, United States of America, **3** Carolina Population Center, University of North Carolina at Chapel Hill, Chapel Hill, NC, United States of America, **4** System Stars, LLC, **5** Independent Consultant, Greensboro, NC, United States of America, **6** Community Enrichment Organization, Tarboro, NC, United States of America, **7** Department of Medicine, School of Medicine, University of North Carolina at Chapel Hill, Chapel Hill, NC, United States of America, **8** Department of Social Medicine and Medicine, School of Medicine, University of North Carolina at Chapel Hill, Chapel Hill, NC, United States of America

* leahf@unc.edu

**Data Availability Statement:** All relevant data are within the paper and its Supporting Information files.

## Abstract

Public health scholarship has increasingly called for the use of system science approaches to understand complex problems, including the use of participatory engagement to inform the modeling process. Some system science traditions, specifically system dynamics modeling, have an established participatory practice tradition. Yet, there remains limited guidance on engagement strategies using other modeling approaches like agent-based models. Our objective is to describe how we engaged adolescent youth in co-building an agent-based model about physical activity. Specifically, we aim to describe how we communicated technical aspects of agent-based models, the participatory activities we developed, and the resulting visual diagrams that were produced. We implemented six sessions with nine adolescent participants. To make technical aspects more accessible, we used an analogy that linked core components of agent-based models to elements of storytelling. We also implemented novel, facilitated activities that engaged youth in the development, annotation, and review of graphs over time, geographical maps, and state charts. The process was well-received by the participants and helped inform the basic structure of an agent-based model. The resulting visual diagrams created space for deeper discussion among participants about patterns of daily activity, important places for physical activity, and interactions between social and built environments. This work lays a foundation to develop and refine engagement strategies, especially for translating qualitative insights into quantitative model specifications such as 'decision rules'.

**Funding:** This project was supported by the National Heart Lung and Blood Institute (Grant Number 5K01HL138159; PI Frerichs). It was also supported in part by the National Institutes of Health under award numbers R01AG047869 and K24HL105493. NRS is grateful to the Carolina Population Center for training support (T32 HD091058) and for general support (P2C HD050924). JK and DJD are with System Stars, LLC and an Independent Consultant, respectively. No funding bodies had a role in the design, collection, analysis, interpretation, nor writing of the study. Its contents are the authors' sole responsibility and do not necessarily represent official NIH views. The funders only provided support in the form of salaries for authors and research materials, but did not have any additional role in the study design, data collection and analysis, decision to publish, or preparation of the manuscript. The specific roles of these authors are articulated in the 'author contributions' section.

**Competing interests:** JK is employed by System Stars, LLC and Damie Jackson-Diop is an independent consultant. This does not alter our adherence to PLOS ONE policies on sharing data and materials.

# Introduction

Understanding how to improve behaviors like physical activity is complex. Many levels of influence, including individual motivations, interpersonal social networks, and built environments, interact to shape behavior over time. Scholars have called for the use of systems science and simulation modeling to help manage the complexity of this type of public health issue [1–4]. Systems science is an interdisciplinary field and approach to inquiry that focuses on understanding interrelated and interacting entities that form a unified whole. Simulation modeling, the process of creating and analyzing a digital prototype that emulates a real-life system, is a method often used in systems science. Furthermore, it is recommended that the simulation models are developed with stakeholders [1, 3, 5, 6], which has potential to produce better models, increase the social capital of communities, and improve the chance that a model successfully influences decision making [7].

To our knowledge, there has been limited work to develop and formalize methods that facilitate stakeholder engagement in building agent-based models for public health issues. Once an agent-based model is created, the simulated animations that result can be useful tools for engaging participants. For example, one description of engagement with an agent-based model related to primary care outcomes focused heavily on simulation results from the ultimate model produced [8]. Yet, there is less information about processes for initial structuring and development of agent-based models with stakeholders. One recent study that engaged stakeholders to build an agent-based model around the issue of food insecurity applied scripts from system dynamics modeling [9]. The authors concluded that the scripts were useful in creating conceptual models, but that future work was needed to establish scripts specific to agent-based models due to the vastly different scale of analysis between agent-based and system dynamics models.

In public health, there has been a relatively strong emergence of scholarly literature focused on participatory approaches to building system dynamics models [10–20]. The popularity of participatory system dynamics is likely due, in part, to the field's historical participatory approach to model building [7], which has included the development, documentation, and sharing of best practices for facilitators [21] of group model building activities called "scripts" [22, 23]. While the detailed protocols are an important tool to support group model building in system dynamics, the choice and adaptation of a given script (or set of scripts) for use with certain communities or groups relies on managing the modeling process outputs as *boundary objects*. Boundary objects are the visual representations of the system dynamics model or model elements that result from the engagement process. In system dynamics, boundary objects such as behavior-over-time graphs, causal loop diagrams, and stock and flow diagrams have been considered critical to the process and valuable for their ability to be transformable by all and represent dependencies across stakeholders [24]. The close correspondence between the visual diagramming conventions of system dynamics models and their underlying mathematics has facilitated the management of boundary objects. In contrast, it is less clear how agent-based models can be managed as boundary objects because agent-based models are often defined in lines of software code that can be difficult to directly visualize.

We sought to develop and pilot test new scripted activities specifically for developing agent-based models. Our aim was to develop an approach that would aid participants' understanding of elements of agent-based models that are difficult to visually represent and, where possible, develop new scripts that would result in boundary objects relevant for agent-based models. In this paper, we describe our process and outcomes from piloting a participatory agent-based model building approach alongside adolescent youth and with a focus on physical activity. We engaged youth as our stakeholders for their unique perspective as well as for the potential

additional benefits including improvement in youth's own leadership, personal agency, and collective empowerment.

## Method

The study was reviewed and approved by the institutional review board at the University of North Carolina at Chapel Hill.

### Setting

We conducted this project in one small town (population of 10,844) in eastern North Carolina. African Americans are 48.3%, Non-Hispanic Whites are 44.8%, and Hispanics are 5.1% of the population [25]. The town is situated in a county that has one of the highest unemployment rates in North Carolina and 24.6% of residents live in households with incomes below the federal poverty level [25].

### Modeling team

In addition to three researchers experienced with participatory modeling, our team included community health leaders. One of the community health leaders and collaborator (DS) has been involved in public health research for over 20 years. Her work has included the formation of a county council for adolescent health and she had been involved in a prior system dynamics participatory modeling project [26]. Another community health leader has experience facilitating youth initiatives (DJ) and has designed and implemented youth curricula with an emphasis on mental and behavioral health. Our team also included public health students with a range of experience areas including the development and facilitation of asset mapping and socio-emotional learning curriculum for high school students. Finally, we worked with the youth participants to develop a list of roles and responsibilities that they could assume on a rotating basis (e.g., ice breaker expert–develops ice breakers for each session, feedback queen/king–communicates what is and is not working during the sessions). This was done to provide leadership opportunities for all youth involved as well as foster ownership and cohesiveness among the group.

### Session development

We developed and implemented content for six sessions that were targeted to last between 1–2 hours each. During the sessions, we sought to elicit information from the students on: 1) important physical activity topics to consider, 2) where physical activity takes place, 3) how youths' location and physical activity changes throughout the day, 4) with whom physical activity takes place, and 5) other factors that influence changes in locations and physical activity levels. Although, the majority of the session content was focused on activities to inform an agent-based model, we also introduced youth to concepts of community-engaged research, implemented team and leadership building activities, and initiated discussions about collecting data for a future quantified model.

When building the model structure, we recognized that agent-based models can be difficult to conceptualize [9]. To make the model structure more accessible, we used the concept of storytelling. We linked the core components of agent-based models; 1) the topic of study, 2) agents, 3) properties and rules, 4) environment, and 5) simulation results, to core components of storytelling: 1) the conflict, 2) characters, 3) character development, 4) setting, and 5) the plot. Table 1 highlights each of these linked concepts plus two examples of the storytelling links in application to agent-based models of physical activity. The first example applies the

**Table 1. Example modeling and storytelling linkages.**

| Modeling | Description | Storytelling Link | Extant Literature Example | Case Study Example |
|---|---|---|---|---|
| Topic of study / conflict | Research question, public health issue | Conflict | The impact of crime on African American women's physical activity and obesity | The impact of social and built environment on rural adolescent's physical activity |
| Agents | Simulated individuals in a model | Characters | African American women aged 18–65 living in Washington DC | Adolescents (ages 14–18) living in a largely rural and African American county in Southeastern US |
| Properties and Rules | Properties: Agent characteristics and goals | Character development | Properties: age, height, household location, income, probability of exercise | Properties: age, height and weight, household location, probability of exercise, mood/affective states |
| | Rules: Specifications for agent movement and decision making in a model | | Rules: agents choose where to exercise, and at what intensity and duration to exercise | Rules: agents choose to move to and from locations, interactions with peers and family members influence choices, perceptions of recreational spaces influence choices |
| Environment | Where agents can go in the model | Environment / setting | Washington DC Wards 5, 7, and 8 | Tarboro, NC |
| Model simulation | How the model 'plays out' when run | Plot | As crime was reduced, leisure time physical activity increased, but this depends on how likely women are to exercise initially. | To be determined |

Note: Modeling example derived from [27] Powell-Wiley, T. M., et al. (2017). "Simulating the Impact of Crime on African American Women's Physical Activity and Obesity." Obesity 25(12): 2149–2155.

link to an agent-based model from the extant literature and the second illustrates the application from our pilot study with youth.

The storytelling analogy was introduced in the first session and integrated into all following session materials. More specifically, we began our sessions with the concept of a storytelling 'conflict' and we elicited responses from the participants about what most 'got in the way' or 'helped' them to be physical active, which was used to refine the research questions. We also highlighted that agents in the model were like the characters of a story and used activities throughout the sessions to elicit more information about important qualities and factors that influenced (or were influenced by) their physical activity levels, akin to character development. The model environment was described as a story's setting and participants were led through activities to identify and describe important locations in their lives and for physical activity. Finally, the model simulation was likened to a story's plot. An existing simulation model was used to illustrate how simulation models produce outcomes over time and specifically highlight emergent dynamics of agent-based models.

## Participant recruitment

Our community partner contacted the school district's superintendent and held several meetings also including the high school guidance counselors and principal. These meetings focused on communicating the scope of project, outlining the resources needed to carry out the project (e.g., meeting space in the school), and how recruitment would take place. The high school distributed information about the project via an email to all students with a survey link they could respond to indicating their interest. In addition, our community partner recruited additional youth using convenience sampling strategies. Specifically, she contacted individuals who had been involved with her organizations' other programs and received recommendations from the guidance counselor. To be eligible participants had to be in grades 9–11 and could not have significant cognitive impairments. After students expressed interest, we held a preliminary meeting to explain the scope of the project. The students completed assent forms and their parents/guardians completed and signed informed consent forms as well. Students were provided a $20 cash incentive for each session that they attended.

## Session evaluation

After each session we asked students to respond to a written survey with 4–6 short-answer questions. The questions asked the students to recall what they learned during each session, points of confusion, as well as general feedback and feedback related to a specific scripted activities (e.g., "What, if anything, did you learn about by participating in the physical activity mapping activity?"). After the final session, we also implemented a low-risk, not time intensive debriefing technique called "keep, start, stop" [28, 29]. In sequence, we asked the participants: "what should we keep/start/stop doing?" for future sessions.

## Overview of session scripts and boundary objects

Fig 1 provides a high-level overview of each session and evaluation. Table 2 provides a detailed summary of each session, description of how elements of storytelling were incorporated, and the specific scripts developed and used. Following, we describe four new scripts we created and implemented specifically for agent-based modeling.

**Graphs Over Time script.** The Graphs Over Time script is a well-established component of group model building in system dynamics that we adapted to understand the typical daily patterns of physical activity of youth [33]. The script engages participants in developing plots of one or more variables that capture how the issue of interest and other relevant factors change over time. This activity is typically done in the early stages of a model building effort in order to help better understand and frame the problem. The standard process for this script involves four steps: 1) a facilitator provides an example plot where the x-axis is labeled as "time" and the y-axis is labeled with variable name(s), 2) participants alone, or in small groups, develop their own plots, 3) the participants share their resulting plots with the group, and 4) the facilitator and participants identify themes that emerge from the graphs.

Our goal was to use a similar process to generate and understand plausible patterns of behavior for an agent-based model that could help guide early model calibrations. We followed these same steps with additional annotation exercises (S1 Appendix). A facilitator asked the participants to chart their physical activity level (y-axis) throughout a typical day (x-axis). However, because agent-based models have a focus on social interactions and the environment, we asked participants to further annotate their plots. Specifically, we asked participants to write who they were with (social interactions) and where they were (environment) on sticky notes and overlay these annotations on their plots (Fig 2A). This process created individual plots as boundary objects. Following the session, we created digital graphs that aggregated the groups' plots. Photos of their individual graphs and the aggregated plots (Fig 2B) were presented and discussed at a subsequent session. Showcasing the aggregated results helped the group identify common themes such as sedentary and active times.

**Mapping Important Locations script.** One of the major advantages of agent-based models is the ability to incorporate geographic information such as road networks, locations of

| Session 1 | Session 2 | Session 3 | Session 4 | Session 5 | Session 6 |
|---|---|---|---|---|---|
| Introduction | Graphs Over Time | Review Graphs Over Time<br><br>Mapping Important Locations | Review simple ABM<br><br>State chart review | Interviewing to Understand Decision Rules: Part 1 | Interviewing to Understand Decision Rules: Part 2 |

**Fig 1. Timeline of major model building activities by session.**

**Table 2. Detailed summary of the six agent-based model building sessions.**

| Session Schedule and Topic | Description of Activities | Link to Storytelling | Approximate Duration of Activity | Existing Sources Used or Adapted |
|---|---|---|---|---|
| **Session #1 –Introductions and orientation to agent-based models** | | | | |
| Setting group norms | Participants were provided with several sheets of paper of two different colors and instructed to write their "Hopes" for the project on one color and "Fears" on the other. Participants then shared their hopes and fears and the sheets were taped to the wall. A facilitator identified themes and then guided a conversation about what types of 'ground rules' or norms the group wished to establish in order to help the group achieve their hopes and reduce their fears. These were recorded and reviewed at the beginning of all subsequent sessions. | N/A | 25 minutes | Hopes & Fears Script [30] |
| Orientation project and modeling | A simple simulation model was constructed to specifically highlight emergent dynamics of agent-based models. The model simulated students in a lunchroom and their selection of where to sit. Students chose where to sit based on the proportion of other students who were similar in their preference for Cardi B or Nicki Minaj music. Further, the model was used to illustrate the elements of storytelling. | N/A | 20 minutes | Adapted from Concept Model Group Model Building Script [30] |
| Physical activity barriers/facilitators | The participants were asked to help identify important variables to consider for the model. The participants were asked to brainstorm responses to the prompt, "What are some things that help or make it harder to be physically active?" | Identifying the "Conflict" | 20 minutes | Adaption of Variable Elicitation Script [30] |
| **Session #2 –Understanding roles in research and daily trends in physical activity** | | | | |
| Leadership in research | Participants were introduced to concepts of participatory action research and citizen science. They were oriented to example research projects that have been implemented in communities similar to theirs. | N/A | 30 minutes | N/A |
| Identifying leadership types | Youth were guided through an activity to identify their leadership type and discuss how they can use their strengths to play a role in improving their community's health. | N/A | 20 minutes | Youth Engaged in Leadership and Learning Curricula [31] |
| Physical activity over time | Participants were shown how to draw an example graph that illustrates how a factor or variable changes over time. They were each provided with templates that had an x-axis of time (24-hour day) and y-axis with physical activity level. They were guided to complete and annotate the graph. | Identifying the "Characters", i.e., who are the agents and what are they doing in time and space? | 45 minutes | Graph over Time for ABMs Script (S1 Appendix)– Adapted from Graphs over Time [30] |
| **Session #3 –Understanding the physical activity environment** | | | | |
| Review Graphs over Time | An visualization of the graphs over time (aggregated across all participant responses) were presented and discussed at the beginning of the session. | N/A | 15 minutes | N/A |
| Team-building exercise | Marshmallow structure challenge–the participants were divided into small groups of 3–4 and provided with a package of supplies (dried spaghetti, masking tape, marshmallows). The groups were then given 15 minutes to build the tallest tower possible. At the end, the group reflected on their process, how they worked together, how they used their leadership styles. | N/A | 25 minutes | Leadership Challenge [32] |

(*Continued*)

**Table 2.** (Continued)

| Session Schedule and Topic | Description of Activities | Link to Storytelling | Approximate Duration of Activity | Existing Sources Used or Adapted |
|---|---|---|---|---|
| Mapping the physical activity environment | Participants were guided to identify their homes on a map of their city and then to identify places that encourage physical activity and make it more difficult. Stickers were placed on a large map and the group discussed the most important places. | Identifying the "Setting", i.e., what is the environment that the agents 'live' in? | 60 minutes | New Script—Mapping Important Locations (S1 Appendix) |
| **Session #4 –Putting characters and environment together** | | | | |
| Model Review | A simple model was created and presented at the beginning of the next session that showed the participants movement between their home and school. | N/A | 20 minutes | N/A |
| State Charts | The concept of "state charts" was introduced to the participants. Draft state charts based on prior session activities were presented to the participants. They were asked to trace their potential pathways among the states. A facilitator led a discussion identifying edits and additions to refine the state charts. | Identifying/expanding on "Character Development" | 20 minutes | New Script–State Chart Review (S1 Appendix) |
| **Session #5 –Gaining a deeper understanding of physical activity** | | | | |
| Recap of State Charts | Refined State Charts were reviewed and discussed. | | 15 minutes | |
| Interviewing for decision rules | The participants were introduced to research interview best practices and practiced interviewing each other in order to gain a deeper understanding of their 'decision rules'. Their 'decision rules' were what influences physical activity. After their interviews, participants shared what they learned and a facilitator helped the group identify themes and key 'decision rules'. | Identifying/expanding on "Character Development" | 45 minutes | New Script—Interviewing to Understand Decision Rules (S1 Appendix) |
| **Session #6 –Putting our story together** | | | | |
| Diagramming decision rules | A facilitator re-capped themes from the prior session's interviews and then asked participants to further annotate the state charts with these new themes and insights in mind. | Identifying/expanding on "Character Development" | 45 minutes | New Script—Interviewing to Understand Decision Rules (S1 Appendix) |

specific exposures of interest, and attributes of spatial features. We developed a new script focused on identifying geographic model boundaries, important locations with potential to affect the outcome of interest, and anticipated movement of individuals between various locations (S1 Appendix). For this scripted activity, we obtained a poster-sized map (5x3.5 feet) of the general location of interest (i.e., where the participants lived, socialized, went to school/work) and smaller maps (8.5x11 inch) for each participant. Prior to the session, participants were asked to identify and take pictures of roughly five important places we should include in the model, including places where physical activity happens and places where it does not in their community. Working in small groups, participants were asked to use stickers to locate where their pictures were taken on the map, discuss the locations and their importance, and identify 3–6 locations that the small group felt were most important to their physical activity. Additionally, the participants were guided to annotate their individual maps with examples of how they typically moved from one location to another, highlighting specific routes and modalities of travel (e.g., walk, car). The facilitator also convened the whole group to locate and discuss their prioritized places, travel routes, and modes of travel on the larger map.

This process created individual maps and an annotated group map as boundary objects. Further, following the session, we used the information to create a simple agent-based model that showed the participants' mobility patterns within their geography. The agent-based model

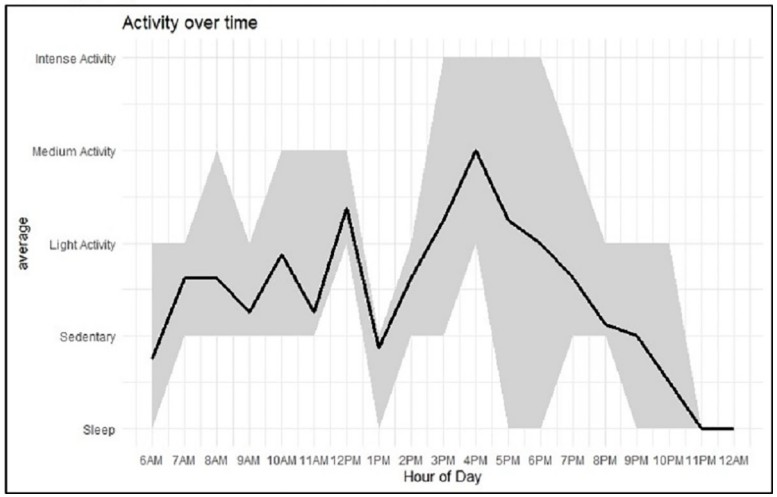

**Fig 2. Resulting artifacts from the Graphs Over Time script.**

was shown to participants and discussed at a subsequent session. This scripted activity helped identify major patterns of mobility among the population of interest and other important concepts that further refined the modeling problem. For example, from this exercise we identified car transportation as the primary mode of travel to and from school.

**State Chart Review script.** The facilitator team used the Graphs Over Time and Mapping Important Locations outputs to develop state charts to include in the model. States and transitions between them are abstractions used in agent-based models to model different conditions that individuals are in at different times (e.g., healthy, infected, dead) and the potential pathways between the states (i.e., an individual can move from healthy to dead or healthy to infected but not dead to healthy). State charts are visual depictions of these conditions. For this scripted activity, we created state charts using AnyLogic (Version 8.4.0 Personal Learning Addition, 2019) and created print-outs of the diagrams. We created states mostly based on the participants' major locations that they identified as part of their typical daily routines (i.e.,

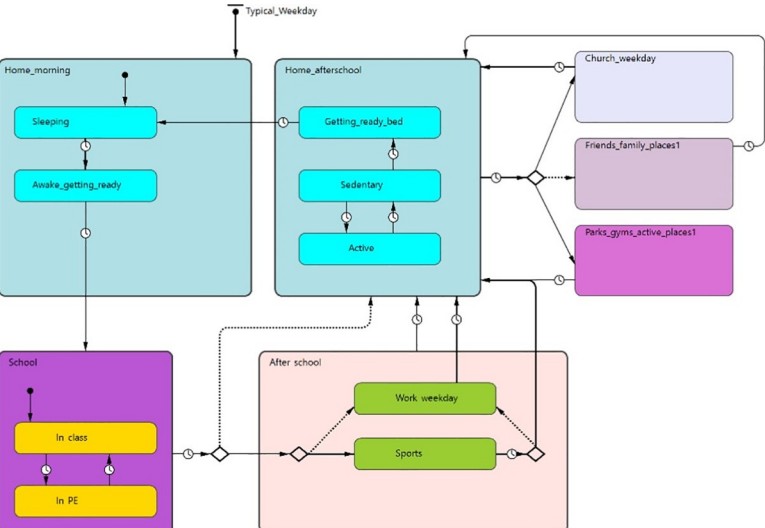

**Fig 3. Resulting artifact from State Chart Review script.**

home, school, and after school/evening places); wherein each location could correspond to a general likelihood of engaging in different levels of sedentary or physical activity. During the session, we presented the state chart and provided print-outs to each participant. The facilitator provided an overview of the state charts and described how they were informed by the prior sessions. Participants were then asked to use highlighters to trace their potential pathways through the states. The group then discussed the comprehensiveness of the states and whether any states or transitions were missing. The state chart was a useful boundary object for discussion and led to the discovery of missing states (i.e., church, family/friend residences) and refinements to the state chart (Fig 3).

**Interviewing to Understand Decision Rules script.** In an agent-based model, decisions and rules need to be formulated to guide each individual agent's transitions between states (e.g., an agent's number of friends who are exercising regularly could influence the probability that they will also exercise). Although the "State Chart Review" activity provided useful information about model states, deeper information surrounding what creates transitions between states was largely missing. Thus, we developed a script to help identify the factors that were perceived as important to creating shifts between states (S1 Appendix).

For this activity, we used a two-phased process implemented across two sessions. During the first session, we guided participants through an activity to interview each other with the goal to stimulate deeper thinking about the issue of interest, physical activity. We introduced them to the basics of interviewing for research (e.g., use of open-ended and probing questions, active listening, reflecting and summarizing answers) and two facilitators role-played a mock interview. Next, we divided the participants into pairs. We provided them with a guiding set of questions and approximately 30 minutes to interview each other. Afterwards, the group convened and each individual shared what they learned from their respective partner about influences on physical activity. A facilitator listened and summarized major themes that emerged from the discussion.

During the second session, the facilitators recapped themes from the prior session's interviews and communicated the session's goal of further annotating state charts. The facilitator projected simplified state charts that focused the participants on the transitions between specific states (i.e., Home not Active, to Home but Active; Home not Active, to Parks/Active Places). Participants were asked to think about what would make them stay versus make the

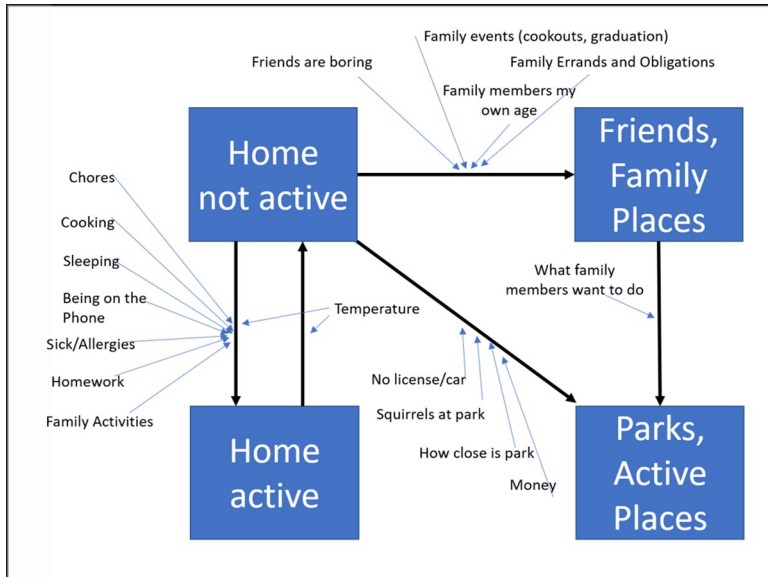

**Fig 4. Resulting example artifact from Interviewing to Understand Decision Rules script.**

various transitions and annotate their individual state charts. Next the facilitator asked each participant, in a round-robin style, to provide factors that influence their transition decisions. This process helped establish additional key relationships and model elements previously missing such as how having family members of similar age influenced activity. This activity resulted in annotated state charts as boundary objects (Fig 4)

## Results

The artifacts from each of the activities provided important insights for different aspects of the model building process. For example, Graphs over Time artifacts revealed insights with implications for the model time scale while Mapping Important Locations provided insights with implications for environmental boundaries and agent-environment interactions within the model. More specifically, the Graphs Over Time artifacts revealed time periods with more (after school) and less (during school) variation in daily activity and Mapping Important Locations artifacts indicated a few key locations (school, home) most relevant to sedentary and physical activity. Furthermore, all artifacts provided insights relevant to data collection and analysis for future model parameterization. For example, the Decision Rules artifacts indicated that social interactions were likely influential in physical activity choices and more information was needed to operationalize the pathways and processes by which friends and family impact activity decisions on a day-to-day basis.

Results from the post-session surveys indicated the participants found the storytelling format and scripted activities useful and rarely had areas of confusion. From the open-ended responses, we found four major themes about what participants learned, specifically participants described: 1) social connections: learning more about and from each other, 2) leadership growth: improving listening and critical thinking skills, 3) spatial knowledge: gaining new skills in reading and understanding maps, 4) physical activity: increasing awareness about their personal physical activity. Specifically, regarding social connections, participants reported learning more about their peers' experiences outside of the study. For example, one participant noted learning that someone in the group had gone surfing and hiking and others

learned that most people enjoyed physical activity with friends and family. In relation to leadership skills, beyond gaining tangible skills in active listening and thinking deeply about themselves, students were also exposed to youth leadership trainings in the area they could attend beyond the lifespan on the project.

From the 'keep, stop, start' activity, the participants indicated that they enjoyed the team-building and leadership activities and suggested they be kept in future sessions. Participants also desired more activities aimed at increasing group cohesion. Suggestions included several outdoor activities (i.e. a picnic and swimming) as well as a trip elsewhere (i.e. to an amusement park or other destination). These findings suggested that the group was continuing to learn more about one another even at the final session and that participants had a desire to form community outside of the traditional meeting space. Participants appreciated the value and insights gained from the model building activities and indicated they would not 'stop' any of them. Suggestions about new things to "start" focused primarily on group logistics such as the meeting room, food, and incentives.

In summary, beyond the tangible spatial and mapping skills as well as critical thinking and listening skills students gained, there was also an appreciation for fostering community both environmentally and on a personal level. What may appear as small details that were not obvious to the success to the study at onset, i.e. food suggestions, building in small celebrations outside of formal meeting spaces, and taking time to get to know one another on a personal level, were crucial elements in the eyes of participants.

## Discussion

We developed and implemented several innovative strategies for partnering with community members to co-develop the basic structure of an agent-based model. Overall, our process was well-received by the participants. Our team found storytelling to be a helpful analogy in communicating the technical aspects of agent-based models. Additionally, the newly scripted activities resulted in visual representations of the model and boundary objects that the participants could view and discuss together.

We found that the storytelling analogy as well as allowing the individual participants to tell their own stories through the "Interviewing for Decision Rules" activity both improved the level of engagement. The use of imagery and narrative is a well-established method in participatory approaches as it relies less on the written word and provides a variety of media that leverages community expertise. A recent study that engaged participants in developing an agent-based model similarly described using aspects of storytelling [34]. The recent study presented clinical case histories, "life stories", to help walk participants through the model structure and logic. In contrast to this method, our team drew on our participant's first-hand knowledge and lived experiences to review state charts, identify pathways, and discuss factors that influence their transitions between places. Our strategy is likely more useful when engaging individuals who have directly experienced the issue at hand; whereas, case histories may be more useful when engaging stakeholders who have second hand experience from individuals they care for whom are affected (e.g., healthcare providers, teachers).

Our process resulted in artifacts that deepened discussion and insights surrounding social interactions and spatial locations, but we also uncovered areas where more work is necessary. Specifically, we found that developing quantifiable decision rules from our resulting artifacts and activities was a challenge. Others have noted that there is limited guidance on the process of translating the visual, qualitative models to quantified simulation models [34]. A recent case study described high-level elements of a participatory approach to develop a dynamic simulation model, but noted that modeling decisions were highly interactive and iterative. Further

research is needed to develop more precise strategies for translating qualitative insights into formulaic and mathematical specifications for agent-based models. We recommend that public health draws from the environmental science discipline, which has a more substantial history of participatory engagement using agent-based models. For example, the field has developed and used "companion modeling" for agent-based modeling that emphasizes the use of role-playing sessions [35–37]. The role-playing sessions use information about potential factors that influence decisions to develop games where participants use spreadsheets or board games to 'simulate' decisions. The role playing sessions can help refine models by identifying irrelevant factors or clarifying relative weights placed on different factors in the decision making process [35, 36]. Alternatively, additional empirical, quantitative research may be necessary for such formulations. As such, one of our project's next steps is to employ primary data collection with the youth and their peers that will aid in model quantification.

In sum, while participatory engagement to develop system dynamics models has been emerging in the public health literature, there have been far fewer examples for agent-based models. Our work adds to the limited guidance for researchers interested in co-developing agent-based models alongside community stakeholders. We found our methods were promising and produced visual elements relevant for agent-based models that participants can equitably own and discuss in the process.

## Supporting information

**S1 Appendix. Agent-based model building scripts.**
(DOCX)

**S1 File.**
(XLSX)

## Acknowledgments

The authors would like to thank the Tarboro High School for their support and assistance with this project.

## Author Contributions

**Conceptualization:** Leah Frerichs, Natalie Smith, Jill A. Kuhlberg, Damie Jackson-Diop, Doris Stith, Giselle Corbie-Smith, Kristen Hassmiller Lich.

**Data curation:** Leah Frerichs, Jill A. Kuhlberg, Gretchen Mason, Damie Jackson-Diop, Doris Stith, Kristen Hassmiller Lich.

**Formal analysis:** Leah Frerichs, Natalie Smith, Jill A. Kuhlberg, Gretchen Mason.

**Funding acquisition:** Leah Frerichs, Doris Stith.

**Investigation:** Leah Frerichs, Natalie Smith, Jill A. Kuhlberg, Gretchen Mason, Damie Jackson-Diop, Doris Stith, Giselle Corbie-Smith.

**Methodology:** Leah Frerichs, Gretchen Mason, Damie Jackson-Diop, Doris Stith, Giselle Corbie-Smith.

**Project administration:** Leah Frerichs, Gretchen Mason, Doris Stith.

**Resources:** Leah Frerichs, Doris Stith, Giselle Corbie-Smith.

**Software:** Leah Frerichs.

**Supervision:** Leah Frerichs, Damie Jackson-Diop, Doris Stith, Giselle Corbie-Smith, Kristen Hassmiller Lich.

**Validation:** Leah Frerichs, Jill A. Kuhlberg.

**Visualization:** Leah Frerichs, Natalie Smith, Jill A. Kuhlberg.

**Writing – original draft:** Leah Frerichs, Natalie Smith, Jill A. Kuhlberg, Gretchen Mason.

**Writing – review & editing:** Leah Frerichs, Natalie Smith, Jill A. Kuhlberg, Gretchen Mason, Damie Jackson-Diop, Doris Stith, Giselle Corbie-Smith, Kristen Hassmiller Lich.

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
