## [Decision Letter · Decision Letter 0]

17 Aug 2020

PONE-D-20-12547

Novel participatory methods for co-building an agent-based model of physical activity with youth

PLOS ONE

Dear Dr. Frerichs,

Thank you for submitting your manuscript to PLOS ONE. After careful consideration, we feel that it has merit but does not fully meet PLOS ONE’s publication criteria as it currently stands. Therefore, we invite you to submit a revised version of the manuscript that addresses the points raised during the review process.

The reviewers have indicated minor changes to the manuscript, as detailed below. The manuscript was an interesting read, and well written, and I look forward to reading the final amended version.

We look forward to receiving your revised manuscript.

Kind regards,

Dr Emma Louise Giles

Academic Editor

PLOS ONE

Journal Requirements:

2.Thank you for stating the following in the Financial Disclosure section:

[This project was supported by the National Heart Lung and Blood Institute (Grant Number 5K01HL138159; PI Frerichs).  It was also supported in part by the National Institutes of Health under award numbers R01AG047869 and K24HL105493.  The funding bodies had no role in the design, collection, analysis, interpretation, nor writing of the study.  Its contents are the authors’ sole responsibility and do not necessarily represent official NIH views.  ].   

We note that one or more of the authors are employed by a commercial company: System Stars, LLC and Independent Consultant

Reviewers' comments:

Reviewer's Responses to Questions

**Comments to the Author**

1. Is the manuscript technically sound, and do the data support the conclusions?

Reviewer #1: Yes

Reviewer #2: Yes

2. Has the statistical analysis been performed appropriately and rigorously? 

Reviewer #1: Yes

Reviewer #2: N/A

3. Have the authors made all data underlying the findings in their manuscript fully available?

Reviewer #1: No

Reviewer #2: Yes

4. Is the manuscript presented in an intelligible fashion and written in standard English?

Reviewer #1: Yes

Reviewer #2: Yes

5. Review Comments to the Author

Reviewer #1: Overall, the manuscript was written well. I would give a suggestion to add a definition for all of the terms that are included in the introduction such as: agent-based models and systems dynamics. Not everyone who reviews will be familiar with these concepts especially understanding the link with CBPR and engagement.

Reviewer #2: This paper seeks to involve youth in the exploration and piloting of scripted activities intended to support the development of an agent-based model surrounding physical activity. The authors implemented a program consisting of six sessions, where students were asked what, when, where, with whom, and trends regarding physical activity via a variety of creative and facilitated activities. After each session, students were asked to indicate what they had learned in the session, as well as for their feedback on the sessions. The authors make a valid case towards developing a model surrounding physical activity that engages youth in the development. The research team was qualified to carry out the work – three researchers experienced with participatory modeling, two experienced community health leaders, and public health students, and youth.

A clear strength of the paper is the well-designed, described, and appended curriculum. Session content is clearly described in a way that others could implement similar activities in their own settings. Methods are clearly aligned with the agent-based model approach. The graphs and plots clearly described and well-selected.

One table that was unclear to me was “Table 1 – Example modeling and storytelling linkages”. Would it be possible to add more detail around how storytelling was incorporated? I did not find this table intuitive to understand after reading the text description. For example, does the content in the table represent hypothetical examples or samples from the project?

The results section is also very short. Is there an opportunity to include a few examples of quotes to support the themes you’re describing? Were there variations among participants in how they experienced the themes you report? Also, the results section only presents the post-session surveys. Consistent with your stated purpose of the paper of piloting new scripted activities for agent-based models, should results also present findings related to the new scripted activities you developed? Are there any insights that the authors/project team gleaned from the artifacts themselves that would be worth reporting on to meet the stated purpose of the paper?

Overall, the manuscript is technically sound and the data support the conclusions. All data is appended, and the authors also include figures and tables to illustrate examples of session artifacts. The authors did a thematic analysis of the summary evaluations, and no statistical analyses were completed. The manuscript is clear and easy to read. One suggestion to further improve clarity is to include definitions of ‘systems science’, ‘simulation modeling’, and ‘agent-based models’ when these terms are introduced early in the paper, to put the methods and findings in context. These terms may not be familiar to the reader.

6. PLOS authors have the option to publish the peer review history of their article (what does this mean?). If published, this will include your full peer review and any attached files.

Reviewer #1: No

Reviewer #2: **Yes: **Katherine Wisener

---

## [Author Response · Author response to Decision Letter 0]

19 Sep 2020

We thank the reviewers for their thoughtful and positive reviews. We addressed each of the reviewers’ minor comments. Please see the responses below. 

Reviewer #1: Overall, the manuscript was written well. I would give a suggestion to add a definition for all of the terms that are included in the introduction such as: agent-based models and systems dynamics. Not everyone who reviews will be familiar with these concepts especially understanding the link with CBPR and engagement.

RESPONSE: We thank the reviewer. We appreciate the suggestion, which was also raised by the other reviewer. We added brief definitions in the introduction to better orient the reader to these terms: 

“Systems science is an interdisciplinary field and approach to inquiry that focuses on understanding interrelated and interacting entities that form a unified whole. Simulation modeling, the process of creating and analyzing a digital prototype to emulate a real-life system, is a method often used in systems science.” 

Reviewer #2: This paper seeks to involve youth in the exploration and piloting of scripted activities intended to support the development of an agent-based model surrounding physical activity. The authors implemented a program consisting of six sessions, where students were asked what, when, where, with whom, and trends regarding physical activity via a variety of creative and facilitated activities. After each session, students were asked to indicate what they had learned in the session, as well as for their feedback on the sessions. The authors make a valid case towards developing a model surrounding physical activity that engages youth in the development. The research team was qualified to carry out the work – three researchers experienced with participatory modeling, two experienced community health leaders, and public health students, and youth. A clear strength of the paper is the well-designed, described, and appended curriculum. Session content is clearly described in a way that others could implement similar activities in their own settings. Methods are clearly aligned with the agent-based model approach. The graphs and plots clearly described and well-selected.

RESPONSE: We thank the reviewer for the thoughtful and positive summary of our research. 

One table that was unclear to me was “Table 1 – Example modeling and storytelling linkages”. Would it be possible to add more detail around how storytelling was incorporated? I did not find this table intuitive to understand after reading the text description. For example, does the content in the table represent hypothetical examples or samples from the project?

RESPONSE: We added a stronger descriptive of how storytelling was incorporated, including a more detailed narrative that provides more context to Table 1:

“…Table 1 highlights each of these linked concepts plus two examples of the storytelling links in application to agent-based models of physical activity. The first example applies the link to an agent-based model from the extant literature and the second illustrates the application from our pilot study with youth. 

The storytelling analogy was introduced in the first session and integrated into all following session materials. More specifically, we began our sessions with the concept of a storytelling ‘conflict’ and we elicited responses from the youth participants about what most ‘got in the way’ or ‘helped’ them to be physical active, which was used to refine the research questions. We also highlighted that agents in the model were like the characters of a story and used activities throughout the sessions to elicit more information from the participants about important qualities and factors that influenced (or were influenced by) their physical activity levels. The model environment was described as a story’s setting and participants were led through activities to identify and describe important locations. Finally, the model simulation was likened to a story’s plot. Finally, an existing simulation model was used to illustrate how simulation models result in outcomes over time and specifically highlight emergent dynamics of agent-based models.” 

The results section is also very short. Is there an opportunity to include a few examples of quotes to support the themes you’re describing? Were there variations among participants in how they experienced the themes you report? 

RESPONSE: Yes, we agree the section was very short. We added details to the section to support the themes as suggested. (p. 17-18) 

Also, the results section only presents the post-session surveys. Consistent with your stated purpose of the paper of piloting new scripted activities for agent-based models, should results also present findings related to the new scripted activities you developed? Are there any insights that the authors/project team gleaned from the artifacts themselves that would be worth reporting on to meet the stated purpose of the paper?

RESPONSE: We thank the reviewer for this suggestion. We added a brief paragraph describing insights from the artifacts: 

“The artifacts from each of the activities provided important insights for different aspects of the model building process. For example, Graphs over Time artifacts revealed insights with implications for the model time scale while Mapping Important Locations provided insights with implications for environmental boundaries and agent-environment interactions within the model. More specifically, the Graphs Over Time artifacts revealed time periods with more (after school) and less (during school) variation in daily activity and Mapping Important Locations artifacts indicated a few key locations (school, home) most relevant to sedentary and physical activity. Furthermore, all artifacts provided insights relevant to data collection and analysis for future model parameterization. For example, the Decision Rules artifacts indicated that social interactions were likely influential in physical activity choices and more information was needed to operationalize the pathways and processes by which friends and family impact activity decisions on a day-to-day basis.” (p. 16-17)

Overall, the manuscript is technically sound and the data support the conclusions. All data is appended, and the authors also include figures and tables to illustrate examples of session artifacts. The authors did a thematic analysis of the summary evaluations, and no statistical analyses were completed. The manuscript is clear and easy to read. 

RESPONSE: Thank you for these positive comments.

One suggestion to further improve clarity is to include definitions of ‘systems science’, ‘simulation modeling’, and ‘agent-based models’ when these terms are introduced early in the paper, to put the methods and findings in context. These terms may not be familiar to the reader.

RESPONSE: We appreciate this suggestion, which was also raised by the other reviewer. We added brief definitions in the introduction to better orient the reader to these terms:

“Systems science is an interdisciplinary field and approach to inquiry that focuses on understanding interrelated and interacting entities that form a unified whole. Simulation modeling is the process of creating and analyzing a digital prototype to emulate a real-life system, which is a tool often used in systems science.”

---

## [Editor Report · Decision Letter 1]

9 Oct 2020

Novel participatory methods for co-building an agent-based model of physical activity with youth

PONE-D-20-12547R1

Dear Dr. Frerichs,

We’re pleased to inform you that your manuscript has been judged scientifically suitable for publication and will be formally accepted for publication once it meets all outstanding technical requirements.

I thank you for revising your manuscript and for considering PLOS ONE.

Kind regards,

Emma Louise Giles

Academic Editor

PLOS ONE
---

## [Editor Report · Acceptance letter]

28 Oct 2020

PONE-D-20-12547R1 

Novel participatory methods for co-building an agent-based model of physical activity with youth 

Dear Dr. Frerichs:

I'm pleased to inform you that your manuscript has been deemed suitable for publication in PLOS ONE. Congratulations! Your manuscript is now with our production department. 

Kind regards, 

on behalf of

Dr. Emma Louise Giles 

Academic Editor

PLOS ONE